# Barriers and facilitators to successful management of type 2 diabetes mellitus in Latin America and the Caribbean: A systematic review

Mar Blasco-Blasco[1], Marta Puig-García[1], Nora Piay[1], Blanca Lumbreras[1,2], Ildefonso Hernández-Aguado[1,2], Lucy Anne Parker[1,2]*

1 Department of Public Health, Universidad Miguel Hernández, Alicante, Spain, 2 CIBER de Epidemiología y Salud Pública (CIBERESP), Madrid, Spain

* lparker@umh.es

## Abstract

**Data Availability Statement:** All relevant data are within the manuscript and its Supporting Information files.

### Background

Given that most evidence-based recommendations for managing type 2 diabetes mellitus (T2DM) are generated in high-income settings, significant challenges for their implementation exist in Latin America and the Caribbean region (LAC), where the rates of T2DM and related mortality are increasing. The aim of this study is to identify the facilitators and barriers to successful management of T2DM in LAC, from the perspectives of patients, their families or caregivers, healthcare professionals, and/or other stakeholders.

### Methods

We conducted a systematic review in MEDLINE, Web of Science, SciELO, and LILACS. We included studies of disease management, prevention of complications and risk factor management. We qualitatively synthesized the verbatim text referring to barriers and/or facilitators of diabetes management according to the Theoretical Domain Framework and described their relative frequencies.

### Findings

We included 60 studies from 1,595 records identified. 54 studies (90%) identified factors related to the environmental context and resources, highlighting the importance of questions related to health care access or lack of resources in the health system, and the environmental context and living conditions of the patients. Issues related to "social influences" (40 studies) and "social/professional role and identity" (37 studies) were also frequently addressed, indicating the negative impact of lack of support from family and friends and clinicians' paternalistic attitude. 25 studies identified patients beliefs as important barriers, identifying issues such as a lack of patients' trust in the effectiveness of the medication and/or the doctor's advice, or preferences for alternative therapies.

**Funding:** This study was funded by a H2020 European Research Council 2018 Starting Grant (Grant number 804761—CEAD). The funder had no role in study design, data collection and analysis, decision to publish, or preparation of the manuscript.

**Competing interests:** The authors have declared that no competing interests exist.

## Conclusions

Successful diabetes management in LAC is highly dependent on factors that are beyond the control of the individual patients. Successful disease control will require emphasis on public policies to reinforce health care access and resources, the promotion of a patient-centred care approach, and health promoting infrastructures at environmental level.

## Introduction

The global burden of diabetes has been continuously increasing in the past decades and this trend is expected to continue in the coming years [1, 2]. As globalization continues and markets for unhealthy commodities expand, individuals living in low- or middle-income countries become increasingly exposed to obesogenic environments making sedentary lifestyles and unhealthy diets and behaviours more frequent [3]. This, combined with the lack of prevention and access to appropriate healthcare, has led to a situation where we now have a higher prevalence of diabetes, more diabetes complications and more deaths due to diabetes in low- or middle-income countries [4]. In 2019, the International Diabetes Federation (IDF) estimated that diabetes affects 463 million people worldwide, with a prevalence of 9.3% among those aged 20 to 79 years, four-fifths of whom live in low- and middle-income countries [5].

In Latin America and the Caribbean (LAC), the lower consumption of fruit and vegetables, as well as the higher intake of saturated fats, sugar and salt in comparison with other regions in the world, is leading to a remarkable problem of obesity and, consequently diabetes [6]. Important inequalities in the distribution of diabetes by gender, ethnicity, education and socio-economic position have been reported [7–9]. Moreover, diabetes is frequently undiagnosed or poorly treated [10], and consequently, there are high rates of diabetic complications [11, 12]. Diabetic retinopathy is especially relevant in LAC, where the prevalence of blindness due to diabetes is higher than in any other region in the world [13].

Many countries in LAC have developed a universal health insurance program. However, the programs tend to be more focused on patient's rehabilitation and treatment, rather than on the prevention or diagnosis phases of disease [14]. Like most regions in the world strengthening the primary care system for improved secondary and tertiary prevention of diabetes, ensuring the disease and its complications are promptly detected and correctly treated, could reduce the consequences of diabetes in the population. However, challenges remain for the effective implementation of evidence-based recommendations and policies in LAC countries. These countries share many of the challenges that high-income countries face in implementing available evidence on prevention/control of T2DM. However, the fact that most recommendations are based on evidence generated in high-income settings [15], together with the cultural and organizational differences between and within LAC, make implementing evidence-based recommendations particularly challenging [16]. Significant barriers to successful implementation of evidence-based strategies are a likely explanation of the increasing prevalence of diabetes complications. It is therefore necessary to address the features of each particular setting, including barriers and/or facilitators, to ensure that evidence-based actions can be successfully implemented.

The reviews that have so far described barriers to T2DM management also come from high-income countries and have mainly focused on individual behaviours rather than considering the patients' environment [17–19]. The implementation of effective interventions to prevent and manage diabetes in LAC could be enabled if appropriate knowledge of the local

contextual and health care related factors is available. Evidence on interactions between access and/or use of health care and sociocultural environment could explain variations T2DM management over the already available knowledge of social inequalities and diabetes [20]. Therefore, we need to address what challenges people with diabetes in LAC face, as well as their living conditions, given that the barriers to diabetes control may depend on both people and their context. This new knowledge will allow a better design of interventions aimed at appropriate diabetes management.

In this systematic review, we evaluate observational studies with qualitative and/or quantitative methodology to identify the barriers or facilitators to the management of T2DM in LAC countries, from the perspectives of patients, their relatives or caregivers, healthcare professionals, and/or any other stakeholders.

## Methods

We conducted a systematic review to identify barriers and/or facilitators in the management of the T2DM disease in LAC countries. We considered T2DM management to include strategies or care protocols for the control of the disease and its risk factors, as well as prevention of complications such as diabetic retinopathy, kidney or heart disease, and diabetic foot. How successful management of the disease was defined depended on the focus of the different studies included. For qualitative studies, it depended on the perceptions of the participants, and how they defined good or poor control of the disease. In quantitative reports the issue was more complex because it was highly influenced by the researcher who designed the data collection tools. In this case, successful care was considered according to measurements such as adherence to care protocols using predefined tools, questionnaires assessing quality of life, or indicators of health care access. We defined facilitator as any factor that supports the management of T2DM and barrier as any factor that limits the disease management. These factors may be socioeconomic, educational, cultural, behavioural, cognitive, structural, or logistical. The source of the data can be patients, patients' families or caregivers, healthcare professionals or any other stakeholder. We recorded the information related to barriers and facilitators as they were reported in the studies. Depending on the perception of the study participants, the same factor may be perceived as a barrier by some stakeholders and a facilitator by others, e.g. religion.

Paired reviewers independently assessed each stage of the review process, from study selection to data synthesis and analysis, and a third reviewer resolved discrepancies. Identification of barriers and facilitators related to disease management was not always explicitly stated as the objective of the research, and the use of different terminology was common. During the abstract review, if it was apparent that barriers or facilitators to diabetes management were reported, we assessed the full text for inclusion.

The protocol of this systematic review was registered on PROSPERO, https://www.crd. york.ac.uk/PROSPERO/, protocol number CRD42019134938 on 22 July 2019 (Protocol is in S1 File).

### Search strategy

We searched MEDLINE via PubMed, Web of Science, SciELO, and LILACS databases up to 16 June 2020. We elaborated a comprehensive search strategy using Medical Subject Headings [Mesh] "Diabetes Mellitus", as well as Text Words such as "Diabetes" AND ("Facilitator" OR "Enabler") AND ("Barrier" OR "Obstacle" OR "Challenge" OR "Difficult*"). We included the condition that the research was carried out at least in one country in the LAC region, according to the World Bank [21]. Therefore, we added the name of these countries in the search. We

also limited studies to humans, and we did not apply time or study design restrictions. The search strategies used for each database are in S1 Text.

## Study selection

We selected studies that: (1) included individuals with laboratory confirmed or self-reported T2DM who lived in LAC countries (2) described facilitators or barriers related to successful management of the disease; (3) reported original data; (4) were written in English, Spanish, French or Portuguese. Studies that included numerous pathologies were included provided that some participants had T2DM and they their results were reported separately. We excluded studies that included solely patients with other diagnoses, or with healthy individuals focussing on primary prevention of T2DM.

## Data extraction

We extracted the information of interest using an ad-hoc checklist that included the following variables:

1. Study characteristics: first author, year of publication, study location, study focus (patient education, health care access, self-care, adherence to medication and/or advice, reduction of complications), study methods (qualitative, quantitative or mixed), and whether or not the study was carried out in an urban or rural setting.

2. Participants characteristics: medical issue (T2DM either alone or with another disease such as hypertension); participant type (patients, relatives, caregivers, healthcare professionals, healthcare directives, stakeholders), number of participants, number of people with diabetes by sex, age of people with diabetes, socioeconomic status of people with diabetes, ethnicity of people with diabetes.

3. The facilitators and/or barriers related to the management of T2DM described in the study to improve diabetes care.

## Quality assessment

To evaluate the external validity and risk of bias of the selected studies, we developed a seven-item checklist using the Newcastle-Ottawa Scale [22] and the JBI Critical Appraisal Checklist for Qualitative Research [23]. The included domains were: (1) the aim of the study is clearly established; (2a) Sample selection clearly defined; (2b) the study sample is likely to be free from selection bias (only quantitative or mixed method studies); (2c) the intentional sample was appropriate according to the aim (only qualitative or mixed method studies); (3) the methods used are appropriate given the study aim; (4) the results are clearly presented and measures have been taken to ensure their validity; (5) the implications of the findings and their applicability to other contexts were discussed. Each item was recorded as Yes, No or Unclear. We piloted the checklist to assess its ability to discriminate the domains of interest and made the necessary modifications. The critical appraisal checklist is available in S2 Text.

## Data synthesis and analysis

We qualitatively synthesized the verbatim text referring to barriers and/or facilitators of diabetes management and described their relative frequencies. We conducted a qualitative analysis of the identified barriers and facilitators according to the Systematic Text Condensation method developed by Malterud [24]. This four-step strategy includes: 1. reading the texts

several times, 2. identifying units of meaning, 3. condensing the information in sub-themes, and 4. synthesizing the sub-themes in themes. The themes were coded in domains according to the Theoretical Domains Framework (TDF) [25, 26]. To ensure rigour, a constant comparison between the condensed information and the original texts were applied up to reaching an agreement between reviewers. The data were analysed using the ATLAS.ti version 8.4 software package.

## Results

### Study selection and characteristics

We initially identified 1,595 records, of which 168 full-text articles were assessed for inclusion and 60 studies were finally included [27–86] (Fig 1). A detailed description of the characteristics is available in S1 Table. Nearly all the studies were published in or after 2005, 33 of them (55.0%) from 2015 (Table 1). Thirty-four studies (56.7%) were written in English, 15 (25%) in Portuguese, and 11 (18.3%) in Spanish.

The studies involved participants from 16 different countries from the LAC region. Nearly half of the studies included participants from Brazil (46.7%), followed by Mexico (20%). Thirty-six studies (55%) used qualitative methods, 17 (28.3%) quantitative methods and 10 (16.7%) mixed methods. Most of the studies were carried out in urban settings (n = 43, 71.7%). Forty-five (75%) studies targeted diabetes alone, and 15 (25%) studies included diabetes with hypertension or other non-communicable diseases. Forty-nine (81.7%) studies analysed barriers and facilitators from the patients' perspective, 19 (31.7%) included the health professionals' perspective, while some studies included other stakeholders, such as family members, caregivers or health managers.

### Quality assessment

The results of the quality assessment are summarised in Table 2 (individual evaluations for each study is available in S2 Table). Among the 60 studies, 55 (91.7%) had a clearly established aim. The sample selection was clearly defined in approximately half of the articles. In 15 (55.5%) of the quantitative studies, we were unable to rule out selection bias in the study sample. In contrast, the intentional samples in 13 (30.2%) of the qualitative studies were judged to be inappropriate according to the study's aim. Most of the studies (42 studies, 70.0%) used appropriate methodology according to the aims they set. Most of the studies (51 studies, 85.0%) presented the results clearly and took steps to ensure their validity, and many of them (39 studies, 65.0%) discussed their implications for other contexts.

The barriers and facilitators described in the studies were classified according to 13 different theoretical domains, which contained a further 29 themes (Table 3).

### "Environmental context and resources" domain

The most common domain was *"environmental context and resources"*, where 54 studies (90%) identified at least one barrier or facilitator [27–34, 36–52, 54–69, 71, 75, 79–86]. Three different themes were identified: health system context, patient context, and environmental context. In relation to the health system context, factors related to the organization of the system play an important role in T2DM management, such as availability of resources, time planning and communication between the different services. Universal coverage, or lack of, (including availability of health insurance) was identified as a barrier (10 studies) and facilitator (3 studies). Regarding patient context, a significant number of studies highlighted the financial situation of the patients, identifying the need for stability as a key facilitator for following diabetes care protocols including meeting appointments, dietary changes, or exercise regimes. However, the demands of being employed were also considered a barrier in 19 studies where the authors

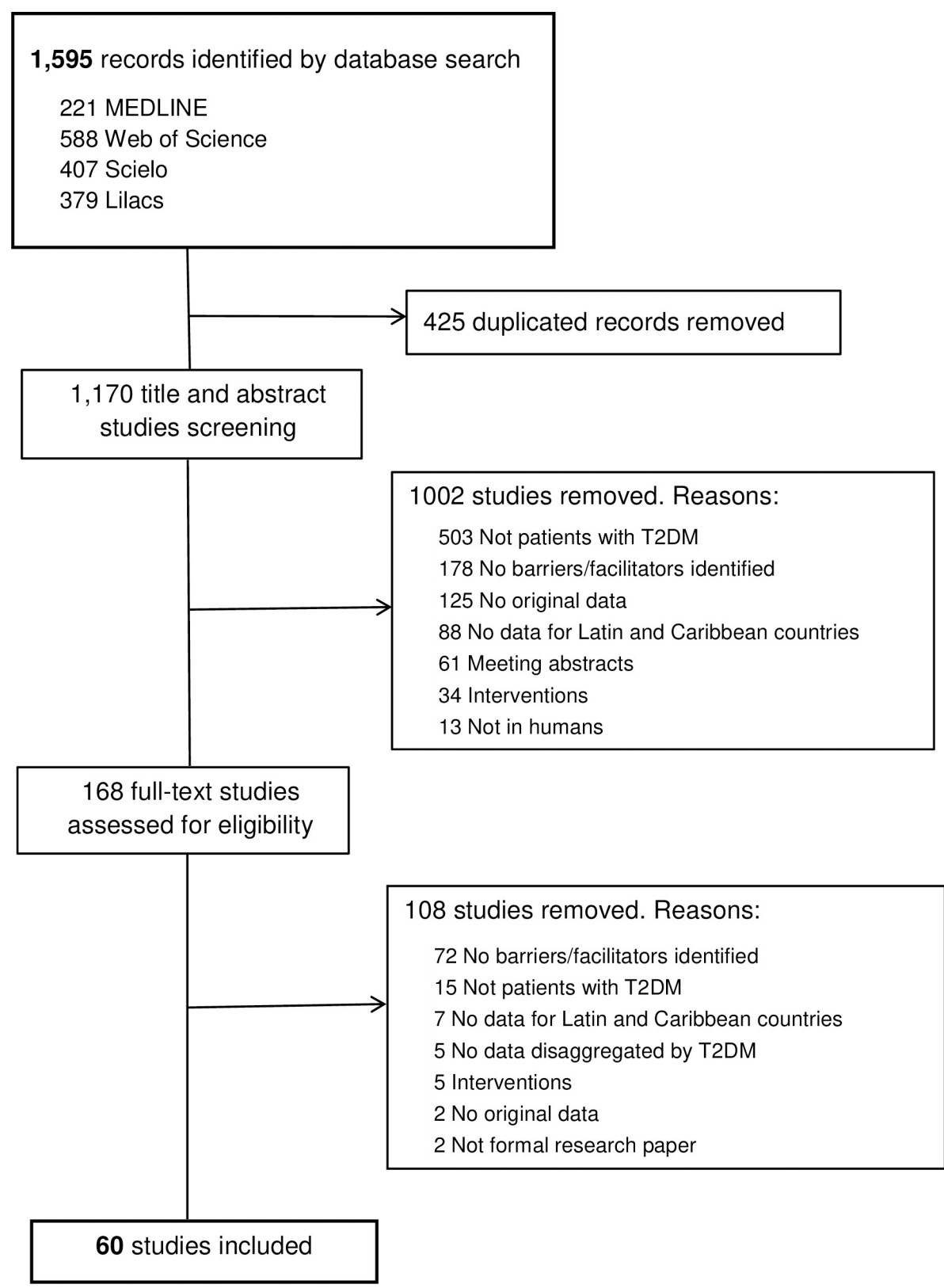

**Fig 1. PRISMA flow diagram.**

**Table 1. Characteristics of sixty studies included.**

| Characteristics | N (%) |
|---|---|
| **Language** | |
| English | 34 (56.7) |
| Portuguese | 15 (25.0) |
| Spanish | 11 (18.3) |
| **Year of publication** | |
| 2015–2020 | 33 (55.0) |
| 2010–2014 | 13 (21.7) |
| 2005–2009 | 13 (21.7) |
| < 2005 (in 1997) | 1 (1.6) |
| **Location[a]** | |
| Brazil | 28 (46.7) |
| Mexico | 12 (20.) |
| Guatemala | 5 (8.3) |
| Peru | 5 (8.3) |
| Barbados | 3 (5.0) |
| Chile | 3 (5.0) |
| Colombia | 2 (3.4) |
| Costa Rica | 2 (3.4) |
| Cuba | 2 (3.4) |
| Other [b] | 7 (11.7) |
| **Study setting** | |
| Urban | 43 (71.7) |
| Rural | 6 (10.0) |
| Urban / Rural | 6 (10.0) |
| Not reported | 5 (8.3) |
| **Study focus[a]** | |
| Patient education | 42 (70.0) |
| Self-care | 40 (66.7) |
| Adherence to medication and/or behaviour change interventions | 28 (46.7) |
| Health care access | 19 (31.7) |
| Reduction of complications | 6 (10.0) |
| **Study method** | |
| Qualitative | 33 (55.0) |
| Quantitative | 17 (28.3) |
| Mixed methods | 10 (16.7) |
| **Health condition studied** | |
| Diabetes mellitus | 45 (75.) |
| Diabetes mellitus plus other non-communicable disease | 15 (25.) |
| **Participant type [a]** | |
| Patients | 49 (81.7) |
| Healthcare professionals | 19 (31.7) |
| Relatives | 4 (6.7) |
| Caregivers | 4 (6.7) |
| Health managers | 2 (3.4) |
| Other stakeholders [c] | 4 (6.7) |

(a) Some studies had more than one location, focus or multiple participant types

(b) Argentina, Belize, Dominican Republic, Ecuador, Jamaica, Trinidad and Tobago, and Venezuela

(c) Other stakeholders: Community members, community leaders, traditional healers.

**Table 2. Quality assessment of the forty-eight studies included.**

| Item | Yes N (%) | No N (%) | Unclear N (%) |
|---|---|---|---|
| Q1 Aim clearly established | 55 (91.7) | 1 (1.7) | 4 (6.7) |
| Q2a Sample selection clearly defined | 29 (48.3) | 14 (23.3) | 17 (28.3) |
| Q2b Randomized or consecutive sample likely to be free from selection bias[a] | 4 (14.8) | 15 (55.5) | 8 (29.6) |
| Q2c Intentional sample was appropriate considering the study's aim[b] | 16 (37.2) | 13 (30.2) | 14 (32.6) |
| Q3 Methods were appropriated to meet study's aim | 42 (70.0) | 6 (10.0) | 11 (18.3) |
| Q4 Clearly presented results, measures taken to ensure their validity | 51 (85.0) | 6 (10.0) | 3 (5.0) |
| Q5 Implications to other contexts discussed | 39 (65.0) | 9 (15.0) | 12 (20.0) |

(a) Only valid for studies which included quantitative methods (n = 27)

(b) Only valid for studies which included qualitative methods (n = 43).

implied that lack of time due to work constraints had a negative impact on following the recommendations or attending the appointments. With regard to the environmental context, studies mentioned the importance of physical assets in the surrounding community like green spaces, sidewalks or cycling lanes to facilitate the behavioural changes recommended in diabetes care. Furthermore, some studies acknowledged that health choices are facilitated by a food environment which promotes access to a balanced and affordable diet in terms of both availability and pricing.

**Table 3. Barriers and facilitators to diabetes care identified in the sixty studies carried out in Latin America and the Caribbean.**

| Domain (N) | Theme | N | Examples of Barriers Identified | N | Example of Facilitators Identified |
|---|---|---|---|---|---|
| **Environmental context and resources [54]** | Health system context | 33 | • Lack of health insurance or health care access <br> • Shortage of physical resources <br> • Lack of human resources <br> • Organizational weaknesses | 18 | • Good insurance coverage and heath access <br> • Strong organizational structure <br> • Multidisciplinary teams <br> • Sufficient human resources |
| | Patient context | 43 | • Financial issues <br> • Work constraints | 6 | • Financial security |
| | Environmental context | 11 | • Weather conditions <br> • Lack of green spaces/ urban infrastructure/ security <br> • Long distance to appointments <br> • Lack of healthy food at workplace | 1 | • Taxing and labelling beverages and food <br> • Providing sidewalks and cycling lanes |
| **Social influences [40]** | (Lack of) Support from family or friends | 22 | • Lack of support related to diet at home and, also, absence of family | 21 | • Support from family / friends to follow diet, translate language, inject insulin, and economical support |
| | Social gatherings | 8 | • Social pressure to disrupt diet <br> • No convenient food or beverages at social gatherings | 1 | • Avoiding social gatherings |
| | Stigma | 7 | • Stigma surrounding illness or use of insulin | 0 | NA |
| | Peer support | 0 | NA | 11 | • Peers support and meeting groups |
| **Social professional role and identity [37]** | Health professionals' role | 27 | • Paternalistic attitude and vertical communication <br> • No patient-centred recommendations | 14 | • Direct communication with patient <br> • Patient-centered recommendations <br> • Relevant educational role of nurses |
| | Patients identity | 12 | • Denial or non-acceptance of the disease | 0 | NA |
| | Gender role of men | 5 | • Prioritising job <br> • Men must be strong <br> • High alcohol and tobacco consumption | 0 | NA |
| | Gender role of women | 8 | • Prioritising taking care of others <br> • Non decision-making power over self-care | 0 | NA |

*(Continued)*

**Table 3.** (Continued)

| Domain (N) | Theme | N | Examples of Barriers Identified | N | Example of Facilitators Identified |
|---|---|---|---|---|---|
| **Knowledge [30]** | Patient knowledge | 18 | • Low health literacy<br>• Bad experience of a family member | 5 | • Mass media providing educational messages<br>• Learning from family experience |
| | Professional knowledge | 9 | • Insufficient knowledge to manage side effects and communicate with patient | 4 | • Updated training provided to health providers |
| **Behavioural regulation [29]** | Following a diet or exercise routine | 20 | • Loss of control on the impulse of eating<br>• Lack of motivation<br>• Diet is monotonous, unfilled, imposed, not fitting preferences and disrupting to daily routine | 0 | NA |
| | Comorbidities and polypharmacy | 14 | • Comorbidities or complications impeding exercise<br>• Vision problems reduce capacity to inject insulin | 0 | NA |
| | Strategies to control glycaemia | 0 | NA | 4 | • Planning daily routine around injecting insulin<br>• Monitoring glycaemia before and after exercise |
| **Emotion [26]** | Emotional burden of disease | 24 | • Fear of side effects, tests and injecting<br>• Depression or stress<br>• Punishment or shame related to insulin<br>• Feeling of loss of independence | 4 | • Fear of death or some complications<br>• No fear of hypoglycaemia<br>• Being calm |
| **Beliefs about consequences [25]** | (Dis) trust | 9 | • Medication, tests and doctor's advice will not work or it is not necessary | 7 | • Trust in medication and doctor advise |
| | Injecting insulin | 5 | • Avoiding starting insulin because it is considered worse<br>• Taking oral medication for granted | 0 | NA |
| | Disease severity | 0 | NA | 6 | • Awareness of disease severity |
| | Home remedies | 11 | • Trust in home remedies as medication substitutes | 3 | • Availability of potentially effective home remedies when economic issues prevent adherence to pharmaceuticals |
| **Reinforcement [15]** | (Lack of) symptoms | 12 | • Side effects of medication, also hypoglycaemia<br>• Absence of symptoms | 11 | • Getting better after following recommendations<br>• Presence of pain or symptoms |
| **Optimism [9]** | Patient faith | 1 | • Faith in God | 6 | • Faith in God<br>• Belief in being cured |
| | Professional attitude | 0 | NA | 2 | • Positive attitude of health professionals |
| **Memory, attention, and decision processes [8]** | Following medication | 8 | • Forgetfulness<br>• Frequent medication or advise changes | 0 | NA |
| **Skills [7]** | Abilities to manage the disease | 7 | • Unable to control diet and cook proper meals<br>• Unable to inject insulin and self-monitoring of blood glucose | 0 | NA |
| **Beliefs about capabilities [6]** | Being capable of controlling the disease | 6 | • Inability to change habits or control food intake<br>• Not injecting insulin correctly | 0 | NA |
| **Intentions [6]** | Changing patient habits | 3 | • No intention of following diet or exercise recommendations | 2 | • Keeping healthy |
| | Professional training | 0 | NA | 1 | • Successfully completing health professionals training |

## "Social influences" domain

Four themes on how *"social influences"* affect diabetes management were identified in 40 studies (66.7%) [27–29, 31, 32, 35, 37, 39–47, 49, 51–53, 57, 59–62, 64–67, 70, 73–77, 79, 81, 83–85]: lack of and support from family and friends, social gatherings, stigma and peer support. More than half of the studies described that family and friends' support is a fundamental piece

in the management of the disease. When this support is lacking, the inclusion in support groups is perceived as a positive facilitator for disease management. However, social gatherings were generally seen as a negative influence because of their role in disrupting the patient's adherence to dietary recommendations. One study described avoidance of these social gatherings as an enabler to disease management, although this could also be a factor related to the social stigma surrounding the illness.

### "Social/professional role and identity" domain

Thirty-seven studies (61.7%) identified questions related to the *"social/professional role and identity"* [27–30, 37–43, 49–53, 55, 57, 59–63, 66, 67, 71–73, 75, 79–86]. Regarding the role of health professionals, several studies indicate that patient-centred recommendations have a positive impact on disease management. Conversely, a more paternalistic approach to doctor-patient communication was identified as a barrier in 21 studies. With regard to the patients, difficulty accepting the diagnosis and identifying as a "patient" was identified in 12 articles as a significant barrier to diabetes care. Barriers associated with the gender roles adopted by men and women were also identified in the studies. For example, prioritising work for men was seen as a factor which impeded diabetes care, while the gender role of women would see them prioritising taking care of others in the family.

### "Knowledge" domain

Regarding *"knowledge"* (30 studies) [27, 29–31, 34, 35, 38, 39, 41, 43, 46, 50–52, 58, 62, 65, 67, 70–73, 75, 80–86], the authors recognised the importance of both patient and professional's knowledge. Lack of professional knowledge was identified as a factor associated with miscommunication and poor management of the side effects. Experiences of family members with diabetes could have a negative influence on disease management in some cases but was also seen as a facilitator in that it could improve the patient's understanding of the disease. Mass media providing educational messages, was reported as a facilitator in 3 studies.

### "Behavioural regulation" domain

This domain appeared in 29 studies [27–29, 31, 32, 37,38, 40, 41, 43, 51, 52, 54, 56, 58, 59, 62, 64, 66, 68, 73, 74, 76, 77, 80, 83–86] and was classified into 3 main themes: (1) Challenges associated with following a diet or exercise routine, where factors such as lack of choice, lack of motivation and monotony were identified; (2) Comorbidities (physical complications such as blindness or diabetic foot) and polypharmacy (adverse effects related with weight gain) could limit follow-up of diet recommendations; and (3) Strategies to control glycaemia, where planning daily activities around the insulin application was seen as a facilitator.

### "Beliefs about consequences" domain

The domain *"beliefs about consequences"* (25 studies) [27, 29–31, 33, 37, 39, 42, 52, 53, 55, 57, 59, 65, 67, 68, 73, 74, 76, 78, 80, 83–86] included studies showing how a lack of trust in the effectiveness of the medication and/or the doctor's advice led to a negation of the treatment, which is an important barrier (9 studies), and was linked to the use of non-therapeutic remedies for glucose control was discussed in 11 studies.

### "Emotion" domain

According to the "*emotion*" domain [27–29, 31, 32, 35, 37–39, 41, 43, 46, 48, 53–55, 66, 67, 70, 73], patients diagnosed with diabetes can feel depression, stress or shame (24 studies).

Nevertheless, for some patients, the fear of death or complications is a strong incentive for treatment adherence.

### "Reinforcement" domain

In the domain "reinforcement" (15 studies) [27, 28, 31, 33, 37, 39, 41, 44, 50, 52, 53, 59, 61, 62, 77, 78, 80, 84, 85], the lack of symptoms and the side effects of medication were seen as factors that could result in people with diabetes stopping the treatment (12 studies).

### Other domains that were less frequently applied

Less frequently applied domains included *"optimism"* where factors such as faith in god, as well as professional and patients' attitudes, were seen to influence diabetes management (9 studies) [27, 37, 41, 46, 55, 67, 71, 80, 85]. Under the domain *"skills"* authors identified how disease management was dependent on the capabilities of the patients related to insulin injection and/or self-monitoring of blood glucose (7 studies) [31, 32, 35, 39, 62, 77, 85]. Along a similar line, regarding *"beliefs about capabilities"* 6 studies identified issues with patients not feeling capable of controlling the disease, for example, not being able to inject insulin correctly [27, 32, 65, 77, 78, 86]. The domain *"memory, attention, and decision processes"* was applied in 8 studies [27, 33, 37, 43, 47, 62, 82, 86], and *"intentions"* (4 studies) [31, 47, 53, 80, 82, 85].

While Table 3 presents the findings of the different stakeholders, supplementary tables shows the findings of the qualitative research according to the participants' perspectives separately (from the perspectives of patients, caregivers, relatives and community members in S3 Table, and from the perspectives of health professionals, health directors and other stakeholders in S4 Table).

## Discussion

We show here that diabetes management in LAC region is a complex issue. Contextual factors appeared to be highly relevant and in some cases individual factors appeared to be on a more secondary plane. The relative importance of contextual factors in diabetes self-management and in diabetic retinopathy screening has been identified in other studies [87, 88].

The most common domain identified was *"environmental context and resources"*, which covers determinants of disease management that lie far beyond the control of the patient. While financial issues and work commitments of the patients were the most frequently identified barrier, factors related to the organization of the health system came closely behind. Shortages of physical and human resources were key barriers to successful diabetes management in the region, as well as environmental determinants such as the lack of safe spaces to do physical exercise or long distances to travel for health care. Even under more individually focused domains such as *"behavioural regulation"*, authors recognise the difficulty to adhere to behaviour change recommendations due to physical limitations associated with complications or co-morbidities, again beyond the scope of individual control. In light of this finding, people with diabetes in LAC region have a limited range of action over their diabetes due to organizational barriers at the level of health system, physical barriers at the level of environment, and socioeconomic barriers at individual level due to social inequalities [89]. According to these results, one would expect that a more rational approach to overcome these barriers should involve both national and local public policies rather than individual interventions to change the behaviour of people with diabetes.

Moreover, how social relationships have been built within society, not only at the patient level but also at health system level, modifies the outcome of the interaction between people and health services. How patients are influenced by social relationships related to family and

friends were key determinants of successful disease management. Following the World Health Organization (WHO) framework for action on the social determinants of health [90], we can also see that numerous factors associated with social stratification such as gender relations or income inequality, educational differences, capacity to understand and apply given recommendations, can lead to increased vulnerability among certain groups and could be key to the inequality in diabetes morbidity and mortality in the region. Even though many of the countries in LAC have universal coverage of primary health care, in practice, out-of-pocket health costs are significant in the region [91]. This could also explain the importance of home remedies for diabetes control which emerged as a key theme in fourteen of the studies.

The effect of the social determinants of health on diabetes management is evidenced also in low-income settings in high-income countries, for instance, in the United States where inequities in health are seriously marked due to unequal distribution of the resources [92, 93]. However, strategies to overcome inequities in health may be different according to the perspective of low- or middle-income countries [94, 95]. Solid evidence shows that a patient-centred care could be useful to integrate family, patients' beliefs and patients' circumstances into diabetes care [96, 97].

Finally, factors at individual level were less represented. We observed two groups of domains, one related to psychological elements such as emotions and beliefs, and another related to the capacity to control disease such as previous knowledge or skills. Despite the fact that it might be desirable to improve the health literacy of people with diabetes to decrease the emotional burden of the disease and increase knowledge, some studies have found that interventions based on increased health literacy are likely to have modest benefits initially on glycaemic control that disappear over time [98, 99].

One of the strengths of this review is the qualitative approach to enhance rigour of the findings. We conducted a constant comparison between condensed information in both subthemes and themes, and coding them within the TDF. In addition, paired reviewers independently reviewed the complete information. Another strength is the wide scope of our review. We included the perspectives of people with diabetes, health professionals, relatives, caregivers and other stakeholders. Furthermore, the barriers and/or facilitators were identified not only at patient level, but also at health system and community level. Another strength of the review search is that it included the main official languages from LAC region and specific databases from this region such as SciELO and LILACS.

One of the limitations of this review is that the TDF might not include all the domains to code the barriers and/or facilitators for successful diabetes management. However, to avoid the exclusion of a new potential domain not included in the TDF, we carried out an open search, but no new domain emerged. An additional limitation of this study is that findings may not applicable to other low-income regions due to the impact of contextual factors. The majority of the studies were carried out urban setting so the challenges faced by individuals living in rural areas are likely to be underrepresented. Furthermore, many of the studies included were from Brazil and Mexico, and other countries were under-represented. This is likely due differences with academic culture and publication of research findings in the different countries in the region. However, we can suppose that the barriers and/or facilitators identified are potentially generalizable to similar sociocultural contexts with limited income and similar health coverage. By synthesising the results of 60 studies from the LAC region, this synthesis of the different types of issues affecting diabetes management can help broaden our understanding of the challenge ahead. The studies were mostly qualitative, and although this type of methodology rarely uses representative samples [100] it is clear that qualitatively synthesised information from the purposively selected participants in a specific setting can generate rich and useful information locally, but it may not be generalisable to the entire LAC region. For

example, qualitative studies describing views and beliefs of specific indigenous groups. Despite this limitation, the review and synthesis of such studies can help shed light on the complexity of the issue in this region, and generate new lines of query for practitioners and researchers in similar contexts.

Approximately 80% of people with diabetes live in low- and middle-income countries, where significant implementation challenges for evidence-based diabetes care exist. Both the IDF and WHO acknowledge that beating diabetes on a global scale will not be achieved by a one-size-fits-all model. Evidence-based recommendations must be adapted to social realities of living each local context. In this review we gain perspective on the complex interactions between the determinants of successful diabetes care in LAC. This implies that time and resources are spent on educating patients may only have a real impact if accompanied with contextual innovations, such as public policies to reinforce infrastructures at social and environmental level to promote healthy behaviours, and improvements in the organization of the health system itself. Taking the complexity of diabetes management into account, approaches to a good control of the disease should include relevant aspects such as public policies to reinforce affordable food, as well as safe infrastructures at environmental level to promote healthy lifestyles [101, 102]. They should also consider strengthening the health system organization to ensure universal coverage is actually achieved in practice and promoting a patient-centred care approach to reduce the effect of social determinants on patient's health.

## Conclusions

In conclusion, successful diabetes management in LAC depends largely on factors that are beyond the control of patients. Although many of the countries in LAC have universal coverage of primary health care, in practice, financial difficulties and work constraints make adherence to diabetes care challenging for patients. Shortages of physical and human resources, as well as environmental determinants such as the lack of safe spaces to do physical exercise or long distances to travel for health care, were also seen as key determinants for successful care. Additionally, we noted that the paternalistic attitude of health care workers, as well as a lack of trust in the effectiveness of the medication and/or the doctor's advice was perceived as an important barrier in many of the studies. In some cases, this mistrust and/or patients' financial constraints prompted individuals to pursue traditional remedies. Furthermore, consideration of how gender roles influence adherence to diabetes and understanding patients' beliefs about medication will be critical.

## Supporting information

**S1 File. Systematic review protocol.**
(DOC)

**S1 Text. Search strategies.**
(DOCX)

**S2 Text. Critical appraisal tool.**
(DOCX)

**S1 Table. Characteristics of the 60 studies included (in detail).** DM: Diabetes Mellitus, T2DM: Type 2 Diabetes Mellitus, HBP: High Blood Pressure, NCC: Neurocysticercosis; NA: Not Applicable, ND: No Data.
(DOCX)

**S2 Table. Critical appraisal assessment in detail.** NA: Not Applicable
(DOCX)

**S3 Table. Barriers and facilitators to diabetes care from the perspective of patients, caregivers, relatives and community members.** NA: Not Applicable
(DOCX)

**S4 Table. Barriers and facilitators to diabetes care from the perspective of the health professionals, health directors and other stakeholders.** NA: Not Applicable
(DOCX)

# Acknowledgments

We would like to thank Dr. Eduardo Bueno Vergara for providing guidance and assistance with the search terms of the literature.

# Author Contributions

**Conceptualization:** Lucy Anne Parker.

**Formal analysis:** Mar Blasco-Blasco, Lucy Anne Parker.

**Funding acquisition:** Lucy Anne Parker.

**Investigation:** Mar Blasco-Blasco, Marta Puig-García, Nora Piay, Blanca Lumbreras, Ildefonso Hernández-Aguado, Lucy Anne Parker.

**Methodology:** Lucy Anne Parker.

**Project administration:** Lucy Anne Parker.

**Writing – original draft:** Mar Blasco-Blasco, Lucy Anne Parker.

**Writing – review & editing:** Mar Blasco-Blasco, Marta Puig-García, Nora Piay, Blanca Lumbreras, Ildefonso Hernández-Aguado, Lucy Anne Parker.

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
