## [Decision Letter · Decision Letter 0]

1 Jun 2020

PONE-D-20-13651

Barriers and facilitators to successful management of type 2 diabetes mellitus in Latin America and the Caribbean: A systematic review

PLOS ONE

Dear Dr. Parker,

Thank you for submitting your manuscript to PLOS ONE. After careful consideration, we feel that it has merit but does not fully meet PLOS ONE’s publication criteria as it currently stands. Therefore, we invite you to submit a revised version of the manuscript that addresses the points raised during the review process.

We look forward to receiving your revised manuscript.

Kind regards,

Cesar Ugarte-Gil, MD, MSc, PhD

Academic Editor

PLOS ONE

Journal Requirements:

2. We note that your search was completed in February 2019.

Please update your search to include studies published in the last 12 months.

Reviewers' comments:

Reviewer's Responses to Questions

**Comments to the Author**

1. Is the manuscript technically sound, and do the data support the conclusions?

Reviewer #1: Partly

Reviewer #2: Yes

2. Has the statistical analysis been performed appropriately and rigorously? 

Reviewer #1: Yes

Reviewer #2: Yes

3. Have the authors made all data underlying the findings in their manuscript fully available?

Reviewer #1: Yes

Reviewer #2: Yes

4. Is the manuscript presented in an intelligible fashion and written in standard English?

Reviewer #1: No

Reviewer #2: Yes

5. Review Comments to the Author

Reviewer #1: Thank you for the opportunity to review this interesting systematic review. The authors studied barriers and facilitators for diabetes care in LAC. The topic is relevant, and certainly the evidence is much needed in this region. The results are very well organised and easy to follow. I have a few suggestions that hopefully will improve this work.

Introduction

1. In the second paragraph, gender, socio-economic, and ethnic inequalities are pinpointed. Although relevant, and much related to the topic, the references used to support these statements are from Europe and US. I will encourage the authors to replace or complement these references with other studies conducted in Latin-America.

Methods

1. In the study selection subheading, it reads that studies were excluded if these did not focus “on patients with an established T2DM diagnosis”. This needs further clarification. Does this mean that only patients with a history of diabetes (self-reported diagnosis) were included? In other words, reports that studied patients with diabetes (self-reported) and people newly diagnosed (blood tests as part of the study) were not included? In this line, how was diabetes history defined? Was this based on self-reported diagnosis only, a combination of self-reported or taking drugs for diabetes, or other means?

Discussion

1. The discussion begins signalling that diabetes is a complex issue and that contextual factors seem to be more relevant than individual profiles. These ideas would benefit from strong references. The findings of this work, although interesting, do not provide evidence to conclude that contextual factors are more important. If there were not any relevant references, these statements should be toned down. This concern would also apply to the first lines of the conclusions; statements appear to be too strong and not fully backed-up by the findings.

2. The limitations of the study argued briefly that qualitative original reports rarely study representative samples. Further discussion on this matter -representative and extrapolation of the results- is warranted. Can the findings of the review really apply to all patients in LAC? Should there be any considerations when interpreting the results across countries (e.g., between Argentina and Mexico)?

3. The conclusion includes arguments about “key determinants for successful care”. This seems quite a strong idea that has not been fully defined nor explored in the results. What is successful care? How is this defined? Does this relate to any care metric (e.g., A1c) or only to the patient’s perspective?

Reviewer #2: This is a well written manuscript that aims to fill an important gap in the literature regarding barriers and facilitators in the management of T2DM in Latin American region. As stated by the authors, this region is experiencing a very large burden of T2DM and it is important to understand the challenges and opportunities uniques to the region.

The study design, a systematic review is a great choice.

Below are concrete comments for authors to consider that I think would improve the paper.

Intro: the study is really about understanding barriers and facilitators of diabetes secondary and tertiary prevention; the authors should consider using these commonly used public health terms.

Per PRISMA guidelines for the reporting of systematic reviews, introduction should include information about the study designs. Were these mostly qualitative studies or also quantitative studies.

Methods:

Study selection: include information about the type of study.

With respect to 2) The objective of the study was to identify or describe facilitators or

barriers related to the management of the disease.

Did the authors rely on the presence of these terms in the stated objectives? If so, please elaborate on the justification for this decision.

Data extraction:

With respect to 3) The facilitators and/or barriers related to the management of T2DM described in the study to improve diabetes care.

This is key given the objective of the study and the statement is vague in light of the types of studies included. Specifically: 1) it is unclear to me whether for quantitative studies, the authors used the results of the studies to identify barriers and facilitators, in other words and for example, did the authors examined results to identify factors associated with a controlled A1c, vs. uncontrolled A1c, first ones were considered facilitators, others barriers. I suspect the authors encountered that type of studies (e.g. registries) and it is important to clarify how they handled/used these results under their own conceptual framework. In fact, some of the characteristics listed in number 2 of the data extraction can fall in the category of barriers (e.g. socioeconomic status). 2) Did they rely on the use of the terms “barriers and facilitators” in the article text. If so, please elaborate on the justification for this decision and consider doing a validation study, examining a random 1-2% of the 65 studies excluded because not focused on “barriers/facilitators”

Did the authors obtain information about the setting: urban, rural, clinic in the hospital, pharmacies, etc.. I think this information would be very valuable to understand the context.

It is recommended to avoid using “diabetic” to refer to patients with diabetes.

Results:

In addition to having table 3 as is, I’d recommend to have 2 similar tables subdivided by type of stakeholder with 2 main groups:

1- Individual patient, caregiver, relative

2- Health care professionals, health managers and other stakeholders.

Figure 1.For the 890 studies removed in early phase, please include the reasons for these exclusions. (as you do for the other exclusions).

6. PLOS authors have the option to publish the peer review history of their article (what does this mean?). If published, this will include your full peer review and any attached files.

Reviewer #1: No

Reviewer #2: Yes: MARIANA LAZO

---

## [Author Response · Author response to Decision Letter 0]

20 Jul 2020

Journal Requirements:

We have checked the manuscript according to the style requirements.

2. We note that your search was completed in February 2019.

Please update your search to include studies published in the last 12 months.

We have updated the search to the 16th June 2020, as requested. 12 new studies have been added to the review. The results section and figure 1 have been updated accordingly. 

Responses to reviewers’ comments

1. Is the manuscript technically sound, and do the data support the conclusions?

Reviewer #1: Partly

Reviewer #2: Yes

We have modified the paper in light of the different comments made and strongly believe that the manuscript has improved.

2. Has the statistical analysis been performed appropriately and rigorously?

Reviewer #1: Yes

Reviewer #2: Yes 

3. Have the authors made all data underlying the findings in their manuscript fully available?

Reviewer #1: Yes

Reviewer #2: Yes 

4. Is the manuscript presented in an intelligible fashion and written in standard English?

Reviewer #1: No

Reviewer #2: Yes 

We have reviewed the language. The final manuscript has been proofread by an experienced native English researcher.

5. Review Comments to the Author

Reviewer #1: Thank you for the opportunity to review this interesting systematic review. The authors studied barriers and facilitators for diabetes care in LAC. The topic is relevant, and certainly the evidence is much needed in this region. The results are very well organised and easy to follow. I have a few suggestions that hopefully will improve this work.

Introduction

1. In the second paragraph, gender, socio-economic, and ethnic inequalities are pinpointed. Although relevant, and much related to the topic, the references used to support these statements are from Europe and US. I will encourage the authors to replace or complement these references with other studies conducted in Latin-America.

Thank you for your comments. We appreciate your suggestions for improving this manuscript. We have replaced the previous references 7 to 9 with studies conducted entirely or partially in LAC region. We have included the following papers: 

● Emmerick ICM, Luiza VL, Camacho LAB, Vialle-Valentin C, Ross-Degnan D. Barriers in household access to medicines for chronic conditions in three Latin American countries. International journal for equity in health. 2015;14(1):115.

● Rivera-Andrade A, Luna MA. Trends and heterogeneity of cardiovascular disease and risk factors across Latin American and Caribbean countries. Progress in cardiovascular diseases. 2014;57(3):276-85.

● Nieblas-Bedolla E, Bream KD, Rollins A, Barg FK. Ongoing challenges in access to diabetes care among the indigenous population: perspectives of individuals living in rural Guatemala. International journal for equity in health. 2019;18(1):1-10.Beagley J, Guariguata L, Weil C, Motala AA. Global estimates of undiagnosed diabetes in adults. Diabetes research and clinical practice. 2014;103(2):150-160.

Methods

1. In the study selection subheading, it reads that studies were excluded if these did not focus “on patients with an established T2DM diagnosis”. This needs further clarification. Does this mean that only patients with a history of diabetes (self-reported diagnosis) were included? In other words, reports that studied patients with diabetes (self-reported) and people newly diagnosed (blood tests as part of the study) were not included? In this line, how was diabetes history defined? Was this based on self-reported diagnosis only, a combination of self-reported or taking drugs for diabetes, or other means?

We agree that the way it was originally worded left this issue unclear. We included studies that were carried out with patients with self-reported type 2 diabetes and/or those with laboratory tests that confirmed their diagnosis. We only excluded studies that included solely patients with other diagnoses, or with healthy individuals focussing on primary prevention of type 2 diabetes. Studies including a mixture of pathologies were included provided that some of the participants had type 2 diabetes, and the results provided the information on type 2 diabetes separately. We have modified the manuscript to make this clearer. This section now reads: “We selected studies that: (1) included individuals with laboratory confirmed or self-reported T2DM who lived in LAC countries (2) described facilitators or barriers related to successful management of the disease; (3) reported original data; (4) were written in English, Spanish, French or Portuguese. Studies that included numerous pathologies were included provided that some participants had T2DM and they their results were reported separately. We excluded studies that included solely patients with other diagnoses, or with healthy individuals focussing on primary prevention of T2DM.”

Discussion

1. The discussion begins signalling that diabetes is a complex issue and that contextual factors seem to be more relevant than individual profiles. These ideas would benefit from strong references. The findings of this work, although interesting, do not provide evidence to conclude that contextual factors are more important. If there were not any relevant references, these statements should be toned down. This concern would also apply to the first lines of the conclusions; statements appear to be too strong and not fully backed-up by the findings.

In light of the comments of the reviewer we have toned down out statement in the conclusions section. Furthermore, we have added 2 references to support the observations made. Please see revised text: “We show here that diabetes management in LAC region is a complex issue. Contextual factors appeared to be highly relevant and in some cases individual factors appeared to be on a more secondary plane. The relative importance of contextual factors in diabetes self-management and in diabetic retinopathy screening has been identified in other studies [75, 76].” 

The references that were added are: 

● Schmidt-Busby J, Wiles J, Exeter D, Kenealy T. Understanding ‘context’ in the self-management of type 2 diabetes with comorbidities: A systematic review and realist evaluation. Diabetes research and clinical practice. 2018;142:321-34.

● Piyasena MMPN, Murthy GVS, Yip JL, Gilbert C, Zuurmond M, Peto T, et al. Systematic review on barriers and enablers for access to diabetic retinopathy screening services in different income settings. PloS one. 2019;14(4). 

2. The limitations of the study argued briefly that qualitative original reports rarely study representative samples. Further discussion on this matter -representative and extrapolation of the results- is warranted. Can the findings of the review really apply to all patients in LAC? Should there be any considerations when interpreting the results across countries (e.g., between Argentina and Mexico)?

The reviewer has made a valid point here. We have added the following text into the discussion/limitations section: “However, we can suppose that the barriers and/or facilitators identified are potentially generalizable to similar sociocultural contexts with limited income and similar health coverage. By synthesising the results of 60 studies from the LAC region, this synthesis of the different types of issues affecting diabetes management can help broaden our understanding of the challenge ahead. The studies were mostly qualitative, and although this type of methodology rarely uses representative samples [100] it is clear that qualitatively synthesised information from the purposively selected participants in a specific setting can generate rich and useful information locally, but it may not be generalisable to the entire LAC region. For example, qualitative studies describing views and beliefs of specific indigenous groups. Despite this limitation, the review and synthesis of such studies can help shed light on the complexity of the issue in this region, and generate new lines of query for practitioners and researchers in similar contexts.” 

We also included some text to acknowledge the bias towards research from certain countries being included: “Furthermore, many of the studies included were from Brazil and Mexico, and other countries were under-represented. This is likely due differences with academic culture and publication of research findings in the different countries in the region.” 

3. The conclusion includes arguments about “key determinants for successful care”. This seems quite a strong idea that has not been fully defined nor explored in the results. What is successful care? How is this defined? Does this relate to any care metric (e.g., A1c) or only to the patient’s perspective?

We agree that this concept is perhaps not well described in the initial version of the manuscript. Our definition of “successful care” depended highly on how it was considered in the manuscripts that were included in the systematic review. Given that many of the included studies were qualitative, the definition of successful care followed what was reported or considered to be successful care according to the perspective of the study participants, ie patient or health care providers etc. In the quantitative studies the idea of what is successful care, would be influenced by choice of outcome measurement and the barriers/facilitators highly dependent on which factors and/or characteristics the researchers who conducted each study chose to measure. In some cases the studies were focussed on a specific issue of diabetes care, such as adherence to medication, in which case care was considered to be successful when individuals were adherent (as per morisky-green or other scale). Similarly, some studies used indicators of glycemic/metabolic control, indicators of health service access, or questionnaires to determine whether the living and health conditions of the patients were good (indicated they were successfully managed) or poor (poorly managed). We have modified the description provided in the methods section to make this more clear. Please see the new description in the methods section: “How successful management of the disease was defined depended on the focus of the different studies included. For qualitative studies, it depended on the perceptions of the participants, and how they defined good or poor control of the disease. In quantitative reports the issue was more complex because it was highly influenced by the researcher who designed the data collection tools. In this case, successful care was considered according to measurements such as adherence to care protocols using predefined tools, questionnaires assessing quality of life, or indicators of health care access.” 

Reviewer #2: This is a well written manuscript that aims to fill an important gap in the literature regarding barriers and facilitators in the management of T2DM in Latin American region. As stated by the authors, this region is experiencing a very large burden of T2DM and it is important to understand the challenges and opportunities uniques to the region.

The study design, a systematic review is a great choice.

Below are concrete comments for authors to consider that I think would improve the paper.

Intro: the study is really about understanding barriers and facilitators of diabetes secondary and tertiary prevention; the authors should consider using these commonly used public health terms.

Firstly, we would like to thank the reviewer for her keen observations. We have used the terms suggested to make the focus of our review clearer. Please see the modification in the introduction section. Inserted text: “Like most regions in the world strengthening the primary care system for improved secondary and tertiary prevention of diabetes, by ensuring the disease and its complications are promptly detected and correctly treated, could reduce the consequences of diabetes in the population.” 

Per PRISMA guidelines for the reporting of systematic reviews, introduction should include information about the study designs. Were these mostly qualitative studies or also quantitative studies.

The reviewer makes a valid point here. This point has now been addressed. Please see the new text: “In this systematic review, we evaluate observational studies with qualitative and/or quantitative methodology to identify the barriers or facilitators to the management of T2DM in LAC countries, from the perspectives of patients, their relatives or caregivers, healthcare professionals, and/or any other stakeholders.”

Methods:

Study selection: include information about the type of study.

With respect to 2) The objective of the study was to identify or describe facilitators or

barriers related to the management of the disease.

Did the authors rely on the presence of these terms in the stated objectives? If so, please elaborate on the justification for this decision.

We included studies where the stated objective clearly indicated that they sought to identify barriers and/or facilitators of diabetes care. However, given that these terms are not always used, during the abstract review if it appeared that determinants of diabetes care were reported in the results section, two researchers independently reviewed the full text and assessed each paper for inclusion according to our inclusion and exclusion criteria. 

We have modified the text in the methods section to make this more clear. Please see text: “Identification of barriers and facilitators related to disease management was not always explicitly stated as the objective of the research, and the use of different terminology was common. During the abstract review, if it was apparent that barriers or facilitators to diabetes management were reported, we assessed the full text for inclusion.”

Data extraction:

With respect to 3) The facilitators and/or barriers related to the management of T2DM described in the study to improve diabetes care.

This is key given the objective of the study and the statement is vague in light of the types of studies included. Specifically: 1) it is unclear to me whether for quantitative studies, the authors used the results of the studies to identify barriers and facilitators, in other words and for example, did the authors examined results to identify factors associated with a controlled A1c, vs. uncontrolled A1c, first ones were considered facilitators, others barriers. I suspect the authors encountered that type of studies (e.g. registries) and it is important to clarify how they handled/used these results under their own conceptual framework. In fact, some of the characteristics listed in number 2 of the data extraction can fall in the category of barriers (e.g. socioeconomic status). 2) Did they rely on the use of the terms “barriers and facilitators” in the article text. If so, please elaborate on the justification for this decision and consider doing a validation study, examining a random 1-2% of the 65 studies excluded because not focused on “barriers/facilitators”

This point was also raised by reviewer 1. In some cases the studies were focussed on a specific issue of diabetes care, such as adherence to medication, in which case care was considered to be successful when individuals were adherent (as per morisky-green or other scale). Similarly, some studies used indicators of glycemic/metabolic control, indicators of health service access, or questionnaires regarding the living and health conditions of the patients. In light of these comments, we have made significant changes in the wording of the methods section to clarify the definitions used and how information was managed. Please see text on lines 90-102: “We considered T2DM management to include strategies or care protocols for the control of the disease and its risk factors, as well as prevention of complications such as diabetic retinopathy, kidney or heart disease, and diabetic foot. How successful management of the disease was defined depended on the focus of the different studies included. For qualitative studies, it depended on the perceptions of the participants, and how they defined good or poor control of the disease. In quantitative reports the issue was more complex because it was highly influenced by the researcher who designed the data collection tools. In this case, successful care was considered according to measurements such as adherence to care protocols using predefined tools, questionnaires assessing quality of life, or indicators of health care access. We defined facilitator as any factor that supports the management of T2DM and barrier as any factor that limits the disease management. These factors may be socioeconomic, educational, cultural, behavioural, cognitive, structural, or logistical. The source of the data can be patients, patients’ families or caregivers, healthcare professionals or any other stakeholder.” 

Furthermore, we have added a clarification about how barriers and facilitators were defined: “Identification of barriers and facilitators related to disease management was not always explicitly stated as the objective of the research, and the use of different terminology was common. During the abstract review, if it was apparent that barriers or facilitators to diabetes management were reported, we assessed the full text for inclusion.”

Did the authors obtain information about the setting: urban, rural, clinic in the hospital, pharmacies, etc.. I think this information would be very valuable to understand the context.

We agree that this point would be very interesting to include. We have integrated this question into the manuscript. Please see table 1, and the comments in the results section lines 205-207: “Most of the studies were carried out in urban settings (n= 43, 71.7%)” . Furthermore we have added this as a limitation in the discussion. See text on lines 606-608: “The majority of the studies were carried out urban setting so the challenges faced by individuals living in rural areas are likely to be underrepresented.”

It is recommended to avoid using “diabetic” to refer to patients with diabetes.

Given. We have changed this term to “people with diabetes” throughout the manuscript.

Results:

In addition to having table 3 as is, I’d recommend to have 2 similar tables subdivided by type of stakeholder with 2 main groups:

1- Individual patient, caregiver, relative

2- Health care professionals, health managers and other stakeholders.

We agree with the reviewer that this would be a very interesting way to review the results. We have considered at length how to make these changes but concluded that with the study, the way it was originally designed, it is not possible to split the table into 2. The main reason is related to the inclusion of quantitative studies. It is difficult to identify who is really identifying the barrier or facilitator because the questionnaires used or the observations made is highly influenced by the researchers and what they consider may be a barrier etc. For example, if we present a table reflecting barriers identified by patients and their family members and include those identified in this manner (because the patients are the ones answering the questionnaire), we would not be properly the views of the patients, but rather the barriers identified by the observation. 

For this reason, we feel that only the qualitative reports truly reflect the perceptions of the participants. As suggested by the reviewer, we have prepared 2 tables according to the perspectives using the qualitative studies. At the moment they are included in the supplement but if the editor wishes we can move them to the main body of the article. 

Figure 1.For the 890 studies removed in early phase, please include the reasons for these exclusions. (as you do for the other exclusions). 

As it is suggested, we have included in Figure 1 the reasons for exclusion of the studies after screening title and/or abstract.

6. PLOS authors have the option to publish the peer review history of their article (what does this mean?). If published, this will include your full peer review and any attached files.

Do you want your identity to be public for this peer review? For information about this choice, including consent withdrawal, please see our Privacy Policy.

Reviewer #1: No

Reviewer #2: Yes: MARIANA LAZO

Thank you for the suggestion. We have used it.

---

## [Editor Report · Decision Letter 1]

29 Jul 2020

Barriers and facilitators to successful management of type 2 diabetes mellitus in Latin America and the Caribbean: A systematic review

PONE-D-20-13651R1

Dear Dr. Parker,

We’re pleased to inform you that your manuscript has been judged scientifically suitable for publication and will be formally accepted for publication once it meets all outstanding technical requirements.

Kind regards,

Cesar Ugarte-Gil, MD MSc PhD

Academic Editor

PLOS ONE
---

## [Editor Report · Acceptance letter]

18 Aug 2020

PONE-D-20-13651R1 

Barriers and facilitators to successful management of type 2 diabetes mellitus in Latin America and the Caribbean: A systematic review 

Dear Dr. Parker:

I'm pleased to inform you that your manuscript has been deemed suitable for publication in PLOS ONE. Congratulations! Your manuscript is now with our production department. 

Kind regards, 

on behalf of

Dr. Cesar Ugarte-Gil 

Academic Editor

PLOS ONE